# Gender Dimensions of Climate Change Adaptation Needs for Smallholder Farmers in the Upper East Region of Ghana

**Mohammed Gali Nuhu [1],\* and Kenichi Matsui [2]**

1   Life and Earth Sciences, Doctoral Program in Environmental Studies, University of Tsukuba,
    Ibaraki 305-8577, Japan
2   Faculty of Life and Environmental Sciences, University of Tsukuba, Ibaraki 305-8577, Japan
\*   Correspondence: s2030293@s.tsukuba.ac.jp

**Abstract:** Gender-specific perceptions and needs are critical to better understanding climate change adaptation capacities of local smallholder communities in Africa and elsewhere. As many rural agricultural practices are heavily impacted by male-centered traditional customs and mores, gender dimensions can determine the level of success for policy interventions. To better understand how and what gender dimensions can be important factors for farmer's climate change adaptations, this study attempts to examine smallholder farmers' adaptation needs and perspectives in Ghana's Upper East Region. A focus group discussion and a questionnaire survey were conducted among 200 smallholder farmers. We found that the female respondents, who mostly belonged to low/middle-income groups, emphasized their urgent need for financial support to improve their income. They needed more farmland as 94% of them had only less than 5 acres to farm. In addition, 91% of the female respondents expressed the importance of being connected to farmers' mutual-help groups to share information about common farming needs. We also found gender-specific needs for extension services, farm inputs, climate information, mechanization, and infrastructure. Regarding access to resources, the women respondents had little interest in investing more in farming as the land they borrowed could be taken away by male owners. The study recommends the need for gender-specific support initiatives that prioritize social protection and gender equality.

**Keywords:** climate change; adaptation needs; gender; smallholder farmers; Ghana

## 1. Introduction

Past studies highlighted that climate change would threaten agricultural production and the livelihoods of poor and marginalized rural communities in Africa [1,2]. Sub-Saharan Africa was identified as one of the most vulnerable regions to climate change partly because 50% of its total workforce depended on basic forms of farming [3]. Subsistence farmers were found to be less likely to adapt to climate change due to a limited access to resources and services [4,5]. This access limitation has long exacerbated food insecurity and social disparities [6,7].

Past climate change adaptation studies also acknowledged that equity and gender disparity, among others, affected climate change vulnerability and adaptability [8–14]. Thompson-Hall et al. [15] emphasized the importance of understanding climate change impacts in the interface between regional social dimensions (e.g., gender, identity, power balance, governance, institutional arrangement) and surrounding ecological, economic, and climate conditions. In northern Ghana, Nyantakyi-Frimpong [16] found that smallholder farmers' loss and damage due to dry spells and flash floods were interconnected to gender disparity, age distribution, marital status, health conditions, dominant religious mores, and poverty. The extent to which these factors affect smallholder's disaster loss and damage depend on physiological, institutional, and socio-cultural settings [17]. Some studies in Ghana showed that preferences for adaptation interventions emerge at the

intersection of gender and livelihoods [11,18]. Codjoe et al. found that, in adopting to drought conditions, women tended to rely on seasonal climate forecasts and post-harvest technologies, whereas men preferred obtaining production technologies [11].

Gender studies shed some light on climate change impacts on food production in northern Ghana [16,19,20]. Some studies identified gender-specific climate change adaptation practices and access to resources [2,21,22]. In Ghana's Upper East Region, female farmers were particularly constrained by a lack of farmland ownership, credit access, irrigation facilities, sufficient information about climate change, and sufficient/high-yielding seeds for planting [2,21].

These studies amply illustrated that more studies on gendered constraints to climate change adaptation among vulnerable communities, especially women, are needed to have a good understanding about climate change adaptation practices and prospects in connection to food production. In particular, smallholder communities are the backbone of the economy in Africa and other developing countries at large.

According to the Least Developed Countries Expert Group (LEG) report [23], climate change adaptation is about reducing vulnerability to existing and future climate risks, whereas climate change vulnerability is largely influenced by people's adaptive capacity. Because of disparities in adaptation capacity, climate threats do not equally affect all persons within a region, community, or even household. The capacity to respond to climate change is constrained by the unequal distribution of rights, resources, and power. This is especially true for smallholder women. On average, women make up 43% of the agricultural labor force in developing countries. In Ghana, women constitute 46.4% of the total labor force in primary agricultural production. The Upper East Region of Ghana has about 80% of women contributing to the agriculture labor force [24–26].

Despite their importance in agricultural activities [24], women generally operate on small-scale farms and are less likely to own property. A lack of ownership means a lack of collateral for accessing credit or loan that can be critical for further farm development investment. Smallholder women cannot receive training or extension services as males predominantly control modern farming practices. These challenges are further exacerbated by some cultural norms [27]. IFAD's studies on gender and land rights in Ghana showed that, although women contribute to 80% of the farm labor force in the Upper East Region, they chronically suffered from limited access to and control over resources such as land [26]. Land inheritance is customarily done through a patrilineal line. Decision making on land are left to male chiefs, elders, family heads, and religious leaders. With the absence of male users, women can gain a temporary access to plots with permission from their husbands. After the husband's death, in the absence of male children, the widow loses her access to land. Single women do not usually have access to land. Women can have access to land if it is remotely located and found to be nutrient poor [26,28]. In Ghana's Upper East Region, this male-dominant land tenure system has been largely unchanged, even though the Ghanaian government has passed laws and implemented policies to promote gender equality for decades [16,29].

Another reason that contributed to gender inequality is related to women's educational opportunities. There is an unwritten social/cultural expectation in which women focus on domestic responsibilities. In Ghana, about 44% of women do not have formal education, compared to about 22% of men. Women's lack of access to information and education can limit their capacity to adapt to climate change impacts, such as seasonal droughts, floods, and insect/disease infections [30,31]. In the Upper East Region, Antwi-Agyei et al. [29] found that women with low education background were more disadvantaged in having access to extension services and information that is important for climate change adaptation.

Furthermore, Ghanaian women are poorly represented in national, regional, and global politics [32]. This disparity is heightened in remote areas, where women rarely participate in decision-making processes [31,33]. The absence of women in decision-making

increases their vulnerability to climate change, since their needs and concerns are not represented and are often inadequately addressed.

The overall objective of this research, therefore, is to respond to the needs of further regional studies with the focus on gender-specific climate change perceptions and adaptation needs among smallholder farmers in the Upper East Region of Ghana. This study contributes to the studies on climate change and livelihoods that tend to focus more on farmers in general without much insight on gender dimensions. The study's findings provide useful entry points for strengthening gender consideration in adaptation planning and implementation.

## 2. Materials and Methods

### 2.1. Study Area

The Upper East Region is located in the northeastern part of Ghana. It lies in semi-arid Guinea and Sudan savannahs with a unimodal rainfall pattern. The wet season occurs from May to October, followed by the long dry season from November to April. The mean annual rainfall ranges from 800 mm to 1100 mm, although the rainfall pattern is erratic, spatially and temporally. The dry season has the cold, dry and dusty harmattan air mass from the Sahara Desert with no rainfall and low relative humidity. The average annual temperature ranges from 14 °C to 35 °C [20].

The population of the Region is estimated to be 1,301,226 (48.5% males and 51.5% females), who are predominantly Muslims [34]. The proportion of male household heads is 72.3%. Households headed by women tend to be excluded from the decision-making process for resource access and community development due to the customary patrilineal local governance system. In this region, women (40.2%) are twice more likely than men (22%) to be home makers. A high proportion of women migrate to urban areas in southern Ghana to work as head porters locally known as "kayayei" to earn a living.

Women in the study area predominantly engage in subsistence rain-fed agriculture [35]. About 90% of farm holdings are less than two hectares in size. Farmers mainly use basic implements like hoe and cutlass to cultivate. In some cases, they use bullocks. Major cereal crops grown in the Region are maize, rice, sorghum, millet, and sweet potato. Cereal crops are mostly intercropped with legumes such as cowpea, soybeans, and groundnuts as sources of proteins and to improve soil fertility through nitrogen fixation. Livestock are also reared in this area. Cattle ownership is considered a measure of wealth and social status. Small ruminants are sold sometimes to pay for leasing farmland [36]. The Ministry of Food and Agriculture recognizes the Region's rain-fed dependent cropping practices as most vulnerable to extreme rainfall variability, floods, drought events, and pest/disease infection [37]. Considering these conditions, we selected the Upper East Region.

### 2.2. Data Collection and Analysis

A mixed method approach was used for this study, including in-person interviews with individual farmers and focus group discussions. We obtained approval and assistance from the Department of Agriculture of the Ghana Ministry of Food and Agriculture in the study area in April 2021. In the initial stage of the research, focus group discussion was conducted to clarify types of climate change-related risks and adaptation needs of farmers in the study area. The information we obtained in this preliminary survey helped us formulate our questionnaire survey. The questionnaire was pre-tested on a sample of smallholder farmers $n = 30$ and then changed in response to the issues highlighted during the pre-testing process.

From the Upper East Region's 15 administrative districts, we chose 4 districts with an estimated population of 338,710 (30%). Then, we approached five communities in each district by applying a simple random sampling method. In the following process, a purposive sampling technique was used to reach 10 respondents in each farming community.

Thanks to dedicated and generous support from agricultural extension service personnel who translated the questionnaire into local languages (Grunne, Frafra, and Kusaal), the response rate was 100% in all four districts. Altogether we obtained valid answers from 200 smallholder farmers. The survey in one community took about one hour on average. The survey form was used to capture the responses. The questionnaire survey was conducted in April and May 2021.

The questions were grouped into two sections. The first section focused on socio-demographic characteristics such as age, gender, education, income, farm size, and farming experience. These characteristics were later used to find correlations to respondents' adaptation needs perceptions that are to be clarified in the second section. Questions in the second section aimed to identify: (1) adaptation needs of the respondents to climate change; (2) gendered roles in resource use; (3) land ownership; and (4) decision-making involvement.

The collected data from the questionnaire were coded and entered into a statistical package for the social science (SPSS version 27) work sheet for analysis. Descriptive statistics in the form of percentages and frequencies were used to show the results.

Based on focus group discussions with farmers, we identified some adaptation needs for our analysis, including extension service, climate information, financial support, farm inputs, crop insurance, irrigation facilities, mechanization, and infrastructure. The needs were presented to the respondents in Likert-scale questions, allowing them to rank their preferences from the most important to the least important, as well as measure the degree of agreement among the respondents. We used Kendall's coefficient of concordance (W) to rank the response. This analytical technique is one of the most commonly used nonparametric methods in assessing the level of agreement among a set of observations by raters. The Kendall's coefficient of concordance(W) is denoted as follows:

$$W = \frac{12S}{P^2(n^3 - n)} - pT \tag{1}$$

Kendall's coefficient of concordance is denoted by W, the number of respondents is denoted by p, the number of needs being ranked is denoted by n, the sum of squares is denoted by S, and the correction factor for tied ranks is denoted by T. In order to test the significance of W, we use the Chi-square ($X^2$) statistics which is given by the formulae:

$$X^2 = p(n - 1)W \tag{2}$$

To identify the varied adaptation needs of smallholder farmers. The null hypothesis (Ho) in this situation was that there is no agreement among the respondent's assessment of their adaptation needs.

We investigated the association between farmers' social status (e.g., gender, income) and their adaptation needs using the Pearson's Chi-square ($X^2$) test, cross tabulations, and correspondence analysis. The responses of male and female farmers to adaptation practices, gendered roles, and access to resources were examined.

## 3. Results and Discussion

### 3.1. Smallholder Farmers' Socio-Demographic Characteristics

The socio-demographic characteristics of the respondents show a gradual trend of aging among farmers in the study area. Their age ranged from 21 to 85 years old, and the mean age was 47 years old. The 20–39 age group consisted of 38% of the total respondents, mostly women (Table 1). Regarding education, 67% of the respondents had no formal or non-formal education. This constituted about 59% of the female respondents and about 41% of the male respondents. The female illiteracy rate of the respondents corroborated with the female illiteracy rate of 59.4% in the Upper East Region [35]. Concerning farming experience, the respondents had an average of 17 years of farming experience. This shows

that their perceptions can be relied on for analysis in this study, given their socio-economic status and farming background.

Regarding their farming, 70.5% of the respondents cultivated less than five acres of farmland while 24.5% cultivated the area between 5 and 10 acres. Only 5% cultivated more than 10 acres. This implies that the respondents were predominantly smallholder farmers.

Considering the daily minimum wage of GH₵12.53 (US$2.07) in Ghana [38], we categorized and analyzed the income situation of the respondents as low-income group (GH₵50–599), middle income (GH₵600–1999), and high income (GH₵2000+). The monthly farm income showed that 82% belonged to the low-income group, 14% belonged to the middle-income group, and 4% had high-income.

**Table 1.** Socio-demographic characteristics of the respondents.

| Social Characteristic | Category | Frequency | (%) | Minimum | Maximum | Mean |
|---|---|---|---|---|---|---|
| Gender | Male | 100 | 50 | | | |
| | Female | 100 | 50 | | | |
| Education | None | 110 | 55 | | | |
| | No formal | 24 | 12 | | | |
| | Basic | 36 | 18 | | | |
| | SHS | 24 | 12 | | | |
| | Tertiary | 6 | 3 | | | |
| Age | 20–29 | 20 | 10 | | | |
| | 30–39 | 56 | 28 | | | |
| | 40–49 | 56 | 28 | 21 | 85 | 47 |
| | 50–59 | 36 | 18 | | | |
| | 60+ | 32 | 16 | | | |
| Years in farming | 1–10 | 42 | 21 | | | |
| | 11–20 | 80 | 40 | | | |
| | 21–30 | 46 | 23 | 5 | 45 | 17 |
| | 31–40 | 26 | 12 | | | |
| | 41–50 | 6 | 3 | | | |
| Farm size (acres) | <5 | 141 | 70.5 | | | |
| | 5–10 | 49 | 24.5 | 2 | 25 | |
| | 11–20 | 9 | 4.5 | | | |
| | 21–30 | 1 | 0.5 | | | |
| Farm income (monthly) | High income (GH₵2000+) | 8 | 4 | | | |
| | Middle income (GH₵600–1999) | 28 | 14 | | | |
| | Low income (GH₵50–599) | 164 | 82 | | | |

SHS: senior high school; US$1 = GH₵5.90.

### 3.2. Smallholder Farmers' Climate Risk Perceptions

Smallholders must understand risks, priorities, and adaptation needs in order to adequately adapt to climate change [39]. Considering this point, in the second section of the questionnaire survey, we first attempted to identify our respondents' risk perceptions. The results show that about 99% of the respondents had noticed changing weather patterns.

During our focus group discussions, the participants described several climatic hazards, including drought, floods, windstorms, and pest/disease outbreaks. Reflecting on this suggestion, these hazards were presented to our survey respondents and asked to

assess their occurrence and severity on a scale of 1 as rare occurrence, 2 as occasional occurrence, and 3 as frequent occurrence. The result shows that 84.5% of the respondents found erratic rainfalls affected them frequently during the crop season (Figure 1). In connection to this, 75% and 61% observed intermittent/frequent drought and flood incidents, respectively. About 85% of the respondents experienced pest and disease infestation frequently.

The respondents were then asked in a multiple-choice question about how these hazards affected their farming. Their choices included reduced yields due to sporadic rainfall distribution, irrigation water shortage, loss of indigenous varieties, depleting animal feed, and decreasing arable lands. In response, 97.5% chose reduced yields. Other notable impacts were irrigation water shortage (92.5%), loss of indigenous varieties (70.5%), animal feed shortage (60.5%), and decreasing arable lands (55.5%). These results support the findings of Afriyie et al. [40] and Zakaria and Matsui [41], which indicated that smallholder farmers in Ghana had experienced the negative impacts of extreme weather events on food production.

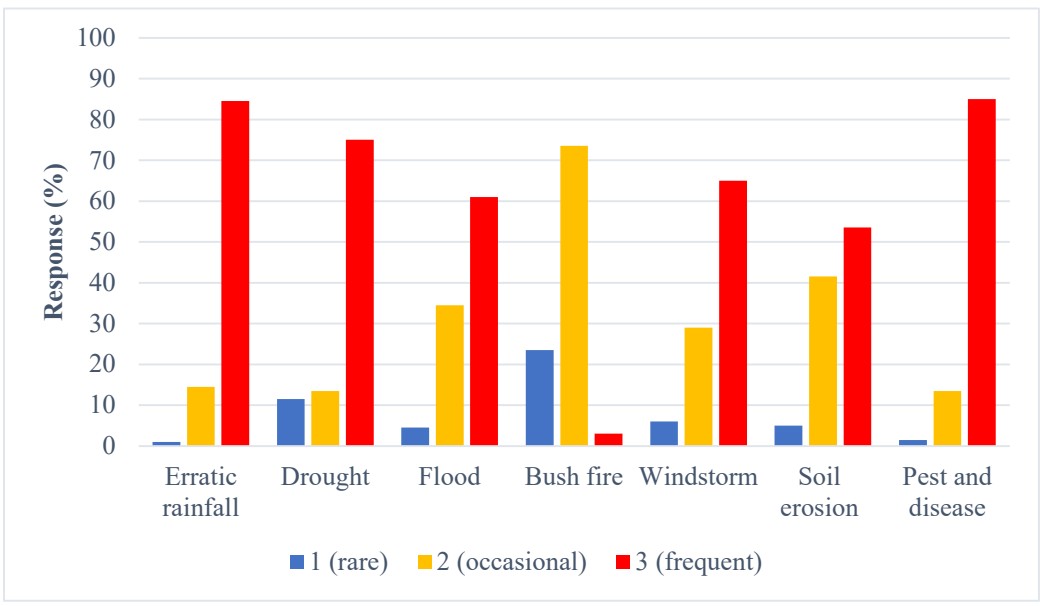

**Figure 1.** Rate of occurrence of climatic hazards. Source: Author's construct, based on survey.

How would you rate the occurrence of the following hazards as a result of the weather changes for the past decade? Please rate occurrence from scale 1 to 3. (Occurrence: 1 = rare, 2 = occasional, 3 = frequent).

### 3.3. Smallholder Farmers Adaptation Practices and Needs by Gender

In the next section of the questionnaire survey, the respondents were asked to indicate adaptation strategies they had applied in response to the climate change-induced hazards mentioned above. About 98% of the respondents adopted some form of adaptation strategies. These include the adoption of early maturing varieties, changing sowing and harvesting time, and soil fertility management. These strategies are largely influenced by gender [18,39]. The chi square $\chi^2$ analysis proved a significant relationship ($p < 0.05$) between gender and adaptation practices (Table 2). Our findings revealed that female smallholder farmers adopted differently when it comes to such adaptation strategies as changing cropping pattens, adopting early maturing varieties, soil fertility management, and livelihood diversification. Some women interplanted cereals with leguminous crops and used animal manure to improve the fertility of their marginalized lands. They also relied on other sources of income to sustain their households, including shea butter

processing, charcoal production and firewood sales, pito (local beer) brewing, soap making, and smock weaving.

We then ranked adaptation needs to identify the priority of their demand (Table 3). Access to extension services, financial support, access to inputs, access to climate information, irrigation, mechanization services, and market facilities were identified as the most important adaptation needs. Kendall's coefficient of concordance (W) was 0.401 with 1% significance. This implies that about 40% of the respondents agreed with the gendered ranking outcome.

**Table 2.** Gender and smallholder farmers adaptation practices cross-tabulation.

| Adaptation Practices by Gender | Male % | Female % | $\chi^2$-Value | Significance Level |
|---|---|---|---|---|
| Changed cropping patterns | 76 | 95 | 14.559 | 0.001 |
| Changed sowing and harvesting time | 97 | 93 | 1.684 | 0.194 |
| Adopted early maturing variety | 96 | 76 | 16.611 | 0.001 |
| Soil fertility management | 69 | 96 | 25.247 | 0.001 |
| Pest and disease management | 47 | 32 | 4.708 | 0.030 |
| Livelihood diversification | 64 | 87 | 14.299 | 0.001 |

(df = 1, Significant at $p < 0.05$).

**Table 3.** Ranking of smallholder farmers adaptation needs.

| Key Adaptation Needs/Priorities | Mean | Ranking |
|---|---|---|
| Extension service | 7.10 | 1st |
| Financial support (Credit/loan) | 6.22 | 2nd |
| Access to inputs | 4.64 | 3rd |
| Access to climate information | 4.48 | 4th |
| Access to irrigation | 4.35 | 5th |
| Mechanization services | 3.41 | 6th |
| Infrastructure (storage facilities, market space) | 3.29 | 7th |
| Crop insurance | 2.52 | 8th |

($n$ = 200, Kendall's W[a] = 0.401, Chi-Square = 561.658, df = 7, Asymp. Sig = 0.000).

### 3.4. Gendered Adaptation Needs of Smallholder Farmers

In this section, the correspondence analysis was used to assess the underlying variables that influenced different adaptation needs by gender (Figure 2). It revealed that the two dimensions explained the differences in adaptation needs and social status. Dimension 1 (72.9% of variance) showed the relevance and impact of social status, such as gender and income, on respondents' adaptation needs preferences. Dimension 2 (17.7% of variance) showed the range of adaptation needs for livelihood security, from the least important to the most important.

Our analysis of these two dimensions showed that the female respondents in low/middle income groups emphasized the need for financial support. The male respondents in the middle-income group needed access to irrigation facilities, infrastructure, and mechanization. Regarding types of services from extension services, the female respondents needed educational/training opportunities, whereas the male respondents needed technological expertise. The need for farm input was found to be associated with the low-income group, especially the female respondents. This may be due to their limited access to credits and productive lands [42,43].

Regarding climate information that is important in supporting adaptation, we found that the female respondents were more likely to show interest in obtaining climate information, but they also showed concerns over a lack of timely information (94%),

information that is difficult to understand (85%), and language barriers in communicating about the climate (97%).

The difference between men and women in terms of crop insurance availability was not substantial (29% and 25%, respectively). This may be due to overall financial constraints and affordability of insurance premium for smallholder farmers in general in the study area. The lack of awareness, understanding, knowledge, and accessibility also affected the adoption of insurance.

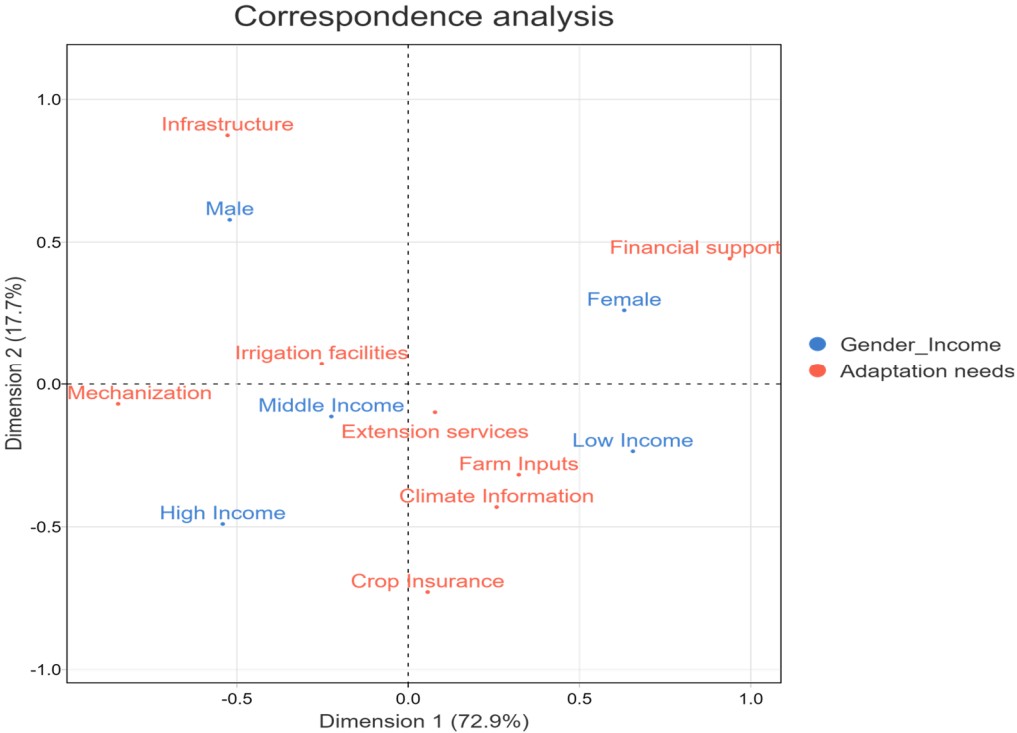

**Figure 2.** Perceptual map of correspondence analysis showing the relationship between adaptation needs and social status (gender and income). Source: Authors' survey.

*3.5. Gender Roles and Access to Resources in Adapting to Climate Change*

In our focus group discussion, we found that farmers were concerned about land acquisition, land preparation, farm maintenance, harvesting, post-harvesting, and marketing. In our questionnaire survey, we highlighted these activities and asked Likert-scale questions (1 = strongly disagree and 5 = strongly agree). The results showed that 91% of the respondents strongly agreed that women were partially involved in land acquisition and preparation. About 85% of the respondents agreed that farming practices such as sowing, weeding, and fertilizer application were jointly carried out by men and women. Nealy all the respondents (97%) strongly agreed that women were dominant in harvesting, post-harvesting activities, and marketing.

Regarding women's access to resources, we found in our focus group discussion that farmers in the study area needed more land, agricultural input, credit, production technology, storage, and marketing facility. At the same time, we found that, with the exception of women in the Tindana family (priests of the earth), women were customarily prohibited from owning land. Women can only have access to the farmland allocated by the extended family.

Considering these situations, we analyzed our questionnaire survey results by correlating land ownership with gender. The result showed that women (73%) relied on land allocation/gift and lease (96%). We found that only men acquired farmland through inheritance (100%). This means that women had very little control over farmland ownership. This is due to patriarchal cultural norms that recognize only men to inherit and own

land through lineage. Clan heads are the sole authority to make decisions about land ownership transitions [27,44].

Understanding these socio-cultural constraints for women, we then tried to understand how women famers coped with crop production challenges in adapting to climate change. We found that 91% of the female respondents emphasized the importance of farmers' mutual-help groups called susu that often function as a village saving and loan association. These organizations provide cash services and share information with female farmers about common farming needs (Table 4). The female respondents tended to belong to these groups.

**Table 4.** Farmer's membership in mutual-help groups.

| | Gender | Farmer Based Organization (FBO) | Village Savings-Loan Association (VSAL) | Local Cash Service or Susu | None |
|---|---|---|---|---|---|
| % within Category *n* = 100 | Male | 35 | 12 | 13 | 40 |
| | Female | 40 | 28 | 23 | 9 |
| | Total | 75 | 40 | 36 | 49 |
| % by gender *n* = 200 | Male | 45 | 30 | 36 | 80 |
| | Female | 55 | 70 | 63 | 20 |
| | Total | 100% | 100% | 100% | 100% |

Source: Author's construct, based on survey. Note: FBO is a locally organized farmers who share ideas and extension services. VSAL is a self-managed and self-capitalized group that uses member fee to lend money. Susu is a traditional rotating saving group observed widely in Ghana. Group members contribute an equal amount of money either weekly, bi-weekly or monthly. The total contribution is paid to one member of the group on a previously agreed schedule.

## 4. Conclusions

This paper examined gendered responses to climate change adaptation needs and coping strategies among smallholders in Ghana's Upper East Region. We demonstrated that, overall, a large proportion of the respondents identified erratic rainfalls, drought events, and flood incidents as foremost challenges. The respondents also experienced irrigation water shortage, the loss of indigenous varieties, animal feed shortage, and decreasing arable land. Regarding adaptation needs, our correspondence analysis revealed that the female respondents in low- and middle-income groups primarily emphasized their need for financial support (e.g., credit access), whereas the male respondents in the middle-income group needed irrigation facilities, infrastructure, and farm machineries. Regarding types of services, the female respondents needed education/training opportunities, whereas the male respondents needed technological expertise. Both male and female respondents did not show much interest in crop insurance. Regarding coping strategies, the women respondents tended to organize and join mutual help groups called susu, which often functioned as village saving and loan associations. These strategies were necessary as these women had only access to land without ownership. This is largely due to traditional patriarchal practices that discriminated against women.

Given these findings, which amply showed different needs between male and female farmers, there is an urgent need to formulate strategies that accommodate women's specific needs in adapting to climate change challenges. Women's active participation in adaptation policy planning and implementation can be better coordinated by empowering susu members with some subsidies and local governance representation. Agricultural extension services and international food security aid organizations may enhance capacity building and training support, especially by targeting those women without formal education. Adaptation programs and actions should place more emphasis on education/training, disaster preparedness, social welfare, and gender equality. In doing so, it is important to convince male farmers, especially traditional leaders, about the importance of

supporting women to acquire land, input, updated market information, and climate information services.

**Author Contributions:** Conceptualization, M.G.N. and K.M.; methodology, M.G.N. and K.M.; analysis, M.G.N. and K.M.; investigation, M.G.N.; data curation, M.G.N. and K.M.; writing—original draft preparation, M.G.N.; writing—review and editing, K.M.; supervision, K.M. All authors have read and agreed to the published version of the manuscript.

**Funding:** This study was supported by JST- Support for Pioneering Research Initiated by Next Generation (SPRING), Grant Number JPMJSP2124.

**Institutional Review Board Statement:** Not applicable.

**Informed Consent Statement:** In the questionnaire survey, the purpose of the survey was explained to the respondents (smallholder farmers), and the interview was conducted after obtaining their consent.

**Data Availability Statement:** Not applicable.

**Acknowledgments:** We are grateful to smallholder farmers in Bawku Municipality, Kassena Nankana Municipality, Pusiga, and Bolgatanga East districts for their time and co-operation during our questionnaire administration. We also thank Agricultural Extension agents (AEAs) in these districts for their support during the research data collection.

**Conflicts of Interest:** The authors declare no conflict of interest.

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
