# Peer review of "Gender Dimensions of Climate Change Adaptation Needs for Smallholder Farmers in the Upper East Region of Ghana"

_sustainability, doi:10.3390/su141610432_

Round 1

Reviewer 1 Report

Authors had revised the MS. I think the MS can be accepted for publication.

Author Response

Thank you for your feedback.

Reviewer 2 Report

Dear authors,

Thank you for your efforts in revising the manuscript. I believe you did a great job in the revision and the changes alleviated my concerns regarding the manuscript.

Good luck

Author Response

Thank you for your feedback

Reviewer 3 Report

The authors tried to make an article related to the Gender Dimensions of Climate Change Adaptation Needs for Smallholder Farmers in the Upper East Region of Ghana. After I am suggesting a very carefully reading of the INSTRUCTIONS FOR AUTHORS https://www.mdpi.com/journal/sustainability/instructions , please see below few of my concerns regarding this manuscript, which need an extensive revision and restructuring the text. 

1.Lines of the manuscript are not numbered, so is more difficult to clearly indicate the places needing corrections.

2.Abstract is much too long, please see the Instructions for authors (“The abstract should be a total of about 200 words maximum“). 

3.Keywords must be separated by semicolons, not by comma.

4.Aim of the study (mandatory last paragraph of the Introduction section to be easier visible and relevant for those interested) should be about novelty, special aspects, reason for choosing the topic, relevance of the topic (which is very confusing for me) with NO references, not about what the authors have done thorough/during the manuscript. Remove the last paragraph of the actual Introduction section and make a proper AIM of the study which you will add after extending Introduction with the actual 2nd section.

5.Check the Instructions for authors regarding the structure of an ARTICLE, (which are given for observance/to be respected, not being optional) and which the authors have not applied.

6.Actual section 2 belongs to Introduction. After adding it, paste the real AIM of the study.

7.Do not forget renumbering the following sections.

8.Moreover, avoid personal manner of addressing (we, our), and replace it with an impersonal one. The text will sound more professional. Revise the entire manuscript in this regard.

9.Subsection 3.2. Data Collection and Analysis

The authors stated the following:

·       We obtained approval and help from the Ministry of Food and Agriculture in the study area.” Please provide no/date of approval. If only verbal approval, remove the sentence, nobody is interested.

·      “The leading author also has served for this ministry for eight years in the study area and obtained support from his colleagues in collection data.” Remove this sentence. Is not a scientific information who helped whom. 

·      In the initial stage of our research the FGD was conducted to clarify types of risks and adaptation needs of farmers in the study area.”  Which are the types of risks you are talking about and which are the needs of the farmers?

·      “This information helped us formulate our questionnaire survey.” It is not “this” but “these”

·      Page 5. The authors stated “the number of respondents is denoted by P”. Also, in the head of the Table 2, last column, is denoted P-value. I suggest changing in the Table 2 the denotation P-value with (not P, not value, it is obvious it is about a value and usually “p” is the correct denotation). You cannot have same denotation for 2 different things.

10. Please detail and respond to the following questions: who made the surveys? who validated them? there were some collaborations with sociologists, specialists in making such questionnaires? How and where were these questionnaires pre-tested before their application to all respondents? based on which criteria the items were chosen/ how do you have chosen/decided the optimal items? based on which criteria, the respondents were chosen? etc. For a clear methodology in describing such questionnaire, I suggest checking and referring to https://doi.org/10.3390/ijerph182010879

·      The respondent's selection method had to be detailed (maybe a flow chart detailing it will help for a better understanding). Did you get an equal number of female and male respondents by random selection? Try to provide clearer all these info.

11. Table 3. The exact values mentioned after the row about Crop insurance, must be moved from here after the title of the Table, in parenthesis, one after the other. The table will have a better aspect.

12. Figure 2 is blurred. Please replace it with a best quality one.

13. Table 4 provide information that can be included in a simple sentence. Remove the table and transform it in text. It is irrelevant for such information an entire Table.

14. Table 5: Head of the Table is not clear. Moreover, it has empty cells, and an empty line (what for?). For a scientific paper they are not allowed. Complete the empty cells (head of the Table) or merge them where is the case. Format of the Tables must be reshaped in MDPI style, check the Instructions for authors.

15. Strengths and weakness of this study are also missing, as the last paragraph of the actual 5th section (which will be future 4th section). 

16. Conclusions must be shortened. This section must briefly present, usually in a single /unique paragraph the main findings of your research. The excess text can be revised to not be in duplicate with the text inserted in other sections and moved In the Results and Discussion sections.  

Author Response

Point 1: Lines of the manuscript are not numbered, so is more difficult to clearly indicate the places needing corrections.

Response 1: Line numbering has been added to facilitate editing of the manuscript content.

Point 2. The abstract is much too long, please see the Instructions for authors (“The abstract should be a total of about 200 words maximum“). 

 Response 2: The abstract was revised in accordance with the author's guidelines.

Point 3. Keywords must be separated by semicolons, not by comma.

 Response 3: Semicolons have been used to separate keywords

Point 4. The aim of the study (mandatory last paragraph of the Introduction section to be easier visible and relevant for those interested) should be about novelty, special aspects, reason for choosing the topic, relevance of the topic (which is very confusing for me) with NO references, not about what the authors have done thorough/during the manuscript. Remove the last paragraph of the actual Introduction section and make a proper AIM of the study which you will add after extending Introduction with the actual 2nd section.

 Response 4: In the final paragraph, the introduction was edited to clearly state the purpose and significance of the study, followed by the structure of the paper.

Point 5. Check the Instructions for authors regarding the structure of an ARTICLE, (which are given for observance/to be respected, not being optional) and which the authors have not applied.

 Response 5: The Sustainability template was used to organize the structure of the article in accordance with the author's guidelines.

Point 6. Actual section 2 belongs to Introduction. After adding it, paste the real AIM of the study.

 Response 6: Section 2 contributes to the aim of the study by identifying a gap based on an evaluation of the literature. Section 2 is included in the introduction to Justify the significance of the study.

Point 7. Do not forget renumbering the following sections.

 Response 7: Sections have been renumbered

Point 8. Moreover, avoid personal manner of addressing (we, our), and replace it with an impersonal one. The text will sound more professional. Revise the entire manuscript in this regard.

 Response 8: A revision has been made in this regard.

Point 9. Subsection 3.2. Data Collection and Analysis

The authors stated the following:

  • ”We obtained approval and help from the Ministry of Food and Agriculture in the study area.” Please provide no/date of approval. If only verbal approval, remove the sentence, nobody is interested.
  • “The leading author also has served for this ministry for eight years in the study area and obtained support from his colleagues in collection data.” Remove this sentence. Is not a scientific information who helped whom. 
  • “In the initial stage of our research the FGD was conducted to clarify types of risks and adaptation needs of farmers in the study area.”  Which are the types of risks you are talking about and which are the needs of the farmers?
  • “This information helped us formulate our questionnaire survey.” It is not “this” but “these”
  • Page 5. The authors stated “the number of respondents is denoted by P”.Also, in the head of the Table 2, last column, is denoted P-value. I suggest changing in the Table 2 the denotation P-value with p (not P, not value, it is obvious it is about a value and usually “p” is the correct denotation). You cannot have same denotation for 2 different things.

 Response 9: A revision has been made in this regard.

Point 10. Please detail and respond to the following questions: who made the surveys? who validated them? there were some collaborations with sociologists, specialists in making such questionnaires? How and where were these questionnaires pre-tested before their application to all respondents? based on which criteria the items were chosen/ how do you have chosen/decided the optimal items? based on which criteria, the respondents were chosen? etc. For a clear methodology in describing such questionnaire, I suggest checking and referring to https://doi.org/10.3390/ijerph182010879

  • The respondent's selection method had to be detailed (maybe a flow chart detailing it will help for a better understanding). Did you get an equal number of female and male respondents by random selection? Try to provide clearer all these info.

 Response 10: Some more details were provided on the questionnaire design/ survey and data collection.

Point 11. Table 3. The exact values mentioned after the row about Crop insurance, must be moved from here after the title of the Table, in parenthesis, one after the other. The table will have a better aspect.

 Response 11: A revision has been made in this regard.

Point 12. Figure 2 is blurred. Please replace it with a best quality one.

 Response 12: The figure has been replaced.

Point 13. Table 4 provide information that can be included in a simple sentence. Remove the table and transform it in text. It is irrelevant for such information an entire Table.

 Response 13: A revision has been made in this regard.

Point 14. Table 5: Head of the Table is not clear. Moreover, it has empty cells, and an empty line (what for?). For a scientific paper they are not allowed. Complete the empty cells (head of the Table) or merge them where is the case. Format of the Tables must be reshaped in MDPI style, check the Instructions for authors.

 Response 14: The heading of table 5 has been revised. Tables in the manuscript were formatted in the MDPI style.

Point 15. Strengths and weakness of this study are also missing, as the last paragraph of the actual 5th section (which will be future 4th section). 

Response 15: A revision has been made in this regard.

Point 16. Conclusions must be shortened. This section must briefly present, usually in a single /unique paragraph the main findings of your research. The excess text can be revised to not be in duplicate with the text inserted in other sections and moved In the Results and Discussion sections.  

Response 16: The conclusion was revised to include the research findings as well as recommendations for policy and future research.

Round 2

Reviewer 3 Report

The authors made a superficial revision, and responded only to shape request but not regarding the content.

Check my previous report and revise consequently.

Paragraph between L111-115 must be removed. Aim of the study is not about describing what the authors have done in each part of the manuscript. Again, check my previous report.

The following major points have been not addressed at all or not properly addressed (aim of the study): 4, 9, 10, 15, 16.

Table 2. df values being constant/having the same value (1), remove this column and after the title of the Table add (for df=1). It has no reason for keeping such un uninformative / repetitive column

L308. No needing Italics

Table 4 is wrongly edited.

Author Response

Paragraph between L111-115 must be removed. The aim of the study is not about describing what the authors have done in each part of the manuscript. Again, check my previous report.

The paragraph that shows the structure of the paper was removed (L 111-115). The objective and the contribution of the paper to the literature were clearly stated in the last paragraph of the introduction " The study’s findings provide useful entry points for strengthening gender consideration in adaptation planning and implementation.

The following major points have been not addressed at all or not properly addressed (the aim of the study): 4, 9, 10, 15, 16.

Details of the type of risk observed and the adaptation needs of the smallholder farmers were provided in the results and discussion section. Please refer to section 3.2. L 232-237 and figure 1. Also, refer to section 3.4. L287-289 and table 3.

The questionnaire target items were designed based on the focus group discussion with smallholder farmers refer to lines 145-146. The lead author designed the questionnaire and edited by the second author both specialists in the field.

The conclusion was divided into two paragraphs to include the findings of the study and recommendations for policymakers. The authors referred to some published articles in the MDPI Sustainability Journal. 

Table 2. df values being constant/having the same value (1), remove this column and after the title of the Table add (for df=1). It has no reason for keeping such un uninformative/repetitive column

A revision has been made in this regard.

L308. No needing Italics

A revision has been made in this regard.

Table 4 is wrongly edited.

A revision has been made in this regard.

Round 3

Reviewer 3 Report

The authors have addressed my suggestions.

This manuscript is a resubmission of an earlier submission. The following is a list of the peer review reports and author responses from that submission.

Round 1

Reviewer 1 Report

This paper is an interesting case study for analyzing the perceptions and differentiated needs of men and women in adapting to climate change in the Upper East Region of Ghana based on the questionnaire survey method. However, some questions have not been well addressed.

1. the section of Disscussion was not found in this manuscript. So, I suggest that authors should give this section. In particular, authors should compare the results of this study with other work to expand the impact of this study.

2. Results: The section was not well organized. Please re-organize the section of results according to the style of this journal. 

3. Some grammatical errors and irregular English writing can easily be found in the full text. Authors should seek a native speaker to improve them. 

Reviewer 2 Report

Dear authors,

Thank you for your submission.

Reviewer 3 Report

The authors tried to make and article related to the Gender Dimensions of Climate Change Adaptation Needs for Smallholder Farmers in the Upper East Region of Ghana. Please see below few of my concerns regarding this manuscript.

Abstract is much too long, please see the Instructions for authors.

Introduction is too poor.

Aim of the study (last paragraph of the Introduction section) should be about novelty, special aspects, reason for choosing the topic, not about what the authors have done

Actual Section 2 (which is the material in the study) must be included in section 3, and all renamed as 2. Materials and methods. Again, please see the Instructions for Authors (which are given for observance/to be respected, not being optional) and which the authors have not applied.

Tables are not respecting MDPI format.

i.e. Table 4. If 4 columns have the same unit of measure (in this case %), I suggest inserting it in the head of the table, not near each numerical value. Is too repetitive.

The structure of the article is unclear, not respecting the sections requested by the Instructions for authors, for an Original article. Shouldn't Section 5 be included in Results?

Please detail: who made the surveys? who validated them? there were some collaboration with sociologists, specialists in such questionnaires? were these questionnaires pre-tested before their application to all respondents? based on which criteria the items were chosen/ how do you have chosen/decided the optimal items? based on which criteria, the respondents were chosen? etc.

How did you calculate the sample size?
The respondent's selection method had to be detailed. Did you get an equal number of female and male respondents by random selection?
Although the use of a statistical program is mentioned in the material and methods section, it is not described anywhere and no statistical test is used to indicate the correlations / differences between the studied variables. The softs and their variants used for the statistical analysis must be provided.

Strengths and weakness of this study are also missing.

Conclusions must be shortened. This section must briefly present, usually in a single /unique paragraph the main findings of your research.